# Genomic Islands Shape the Genetic Background of Both JP2 and Non-JP2 *Aggregatibacter actinomycetemcomitans*

**DOI:** 10.3390/pathogens11091037

**Published:** 2022-09-13

**Authors:** Weerayuth Kittichotirat, Roger E. Bumgarner, Casey Chen

**Affiliations:** 1Systems Biology and Bioinformatics Research Group, Pilot Plant Development and Training Institute, King Mongkut’s University of Technology Thonburi, Bangkok 10150, Thailand; 2Department of Microbiology, University of Washington, Seattle, WA 98195, USA; 3Department of Endodontics and Periodontics, Herman Ostrow School of Dentistry, University of Southern California, Los Angeles, CA 90089, USA

**Keywords:** *A. actinomycetemcomitans*, genomic islands, virulence genes, microbial evolution

## Abstract

*Aggregatibacter actinomycetemcomitans* is a periodontal pathogen associated with periodontitis. This species exhibits substantial variations in gene content among different isolates and has different virulence potentials. This study examined the distribution of genomic islands and their insert sites among genetically diverse *A. actinomycetemcomitans* strains by comparative genomic analysis. The results showed that some islands, presumably more ancient, were found across all genetic clades of *A. actinomycetemcomitans*. In contrast, other islands were specific to individual clades or a subset of clades and may have been acquired more recently. The islands for the biogenesis of serotype-specific antigens comprise distinct genes located in different loci for serotype a and serotype b–f strains. Islands that encode the same cytolethal distending toxins appear to have been acquired via distinct mechanisms in different loci for clade b/c and for clade a/d/e/f strains. The functions of numerous other islands remain to be elucidated. JP2 strains represent a small branch within clade b, one of the five major genetic clades of *A. actinomycetemcomitans*. In conclusion, the complex process of genomic island acquisition, deletion, and modification is a significant force in the genetic divergence of *A. actinomycetemcomitans*. Assessing the genetic distinctions between JP2 and non-JP2 strains must consider the landscape of genetic variations shaped by evolution.

## 1. Introduction

*Aggregatibacter actinomycetemcomitans* strains can be distinguished as JP2 or non-JP2 strains based on the promoter structures of the leukotoxin operon [1,2]. However, contrasting JP2 and non-JP2 strains leads to an erroneous impression of a comparison of equals. The human strains of *A. actinomycetemcomitans* comprise the following five genetic clades: a/d, e/f, b, c, and e′ [3,4]. The JP2 strains represent a clone within clade b, while the non-JP2 strains are distributed in all recognized clades, including clade b. Notably, the strains of different clades may differ as much as 15% in their genome content [3,4,5]. Therefore, assessing the genetic distinctions between JP2 and non-JP2 strains must consider the landscape of genetic variations shaped by evolution. Details of how these genetic variations were derived and their impact on virulence remain to be elucidated.

In a series of studies, we noted numerous genomic islands in *A. actinomycetemcomitans* strains, irrespective of their designations as JP2 or non-JP2 clones [3,4,5]. We hypothesize that the gain and loss of genomic islands are central to the evolution of *A. actinomycetemcomitans*. To test the hypothesis, this study examined (i) the distribution of genomic islands among *A. actinomycetemcomitans* strains, (ii) the features of island insertion sites, and (iii) the features of islands that carry known virulence determinants of *A. actinomycetemcomitans*.

## 2. Results and Discussion

### 2.1. Features of A. actinomycetemcomitans Genomes

Thirty-one human strains and the serotype b strain RHAA1, isolated from a rhesus monkey [6], were included in this study. The human strains comprise five clades (a/d, e/f, b, c, and e′), as identified in our previous studies [3,4,5]. Strain HK1651 (serotype b) is a JP2 clone. In the following, we will organize the results based on genetic clades whenever possible to avoid confusion. However, the serotype designations of the strains will be used if needed, for example, when the results differ among the strains of the same clades.

A total of 66,205 genes were found in 32 *A. actinomycetemcomitans* genomes and assigned to 4276 distinct homologous gene clusters. The data can be accessed via our online Gene Table Viewer tool at http://expression.washington.edu/genetable/script/gene_table_viewer?organism=aa_ha&build=17_07_20 (accessed on 21 July 2022). Among the homologous gene clusters, 2648 (61.93%) and 1628 (38.07%) were core and variable genes, respectively. The variable genes constitute approximately 12–22% of the total genes and occupy 10–19% of the total bases in the *A. actinomycetemcomitans* genomes (Appendix A). The number of core genes in this analysis is significantly higher than in our previous studies [3,4] because of the more stringent criteria for genomic island identification and the more lenient criteria for defining core genes.

The clade b/c strains from humans were smaller in genome size than other clades (Appendix A), suggesting a genome reduction that accompanied their adaptation to humans. Alternatively, the reduced genome size could be attributed to the loss of natural competence of clade b/c strains [7,8] and the inability of the strains to acquire DNA via horizontal gene transfer, as suggested by Jorth and Whiteley [9].

Appendix A show a summary and the details of 688 genomic islands found in the 32 *A. actinomycetemcomitans* genomes. Specifically, between 3 and 39 genomic islands were identified in each genome. As expected, the poor genome assembly led to lower numbers of identified genomic islands in some strains. These include the following: (i) strain SA508 (549 contigs, 10 genomic islands), (ii) strain SA3733 (787 contigs, 3 genomic islands), (iii) strain SA269 (511 contigs, 8 genomic islands), (iv) strain SA2200 (436 contigs, 18 genomic islands), and (v) strain ANH9776 (590 contigs, 6 genomic islands). The strain D7S-1 has the highest number of genomic islands (a total of 39 genomic islands), occupying 300,564 bases of its genome, followed by the strain RHAA1 (26 genomic islands of 281,742 bp).

### 2.2. Islands May Drive the Evolutionary Divergence of the Species: Distribution of the Homologous Genomic Island Groups among the A. actinomycetemcomitans Strains

Five of the 32 strains were excluded from the analysis due to poor assembly results (strains SA508, SA3733, SA269, SA2200, and ANH9776). Six hundred and forty-three genomic islands were identified in 27 *A. actinomycetemcomitans* genomes, and 581 of these islands were assembled into 56 homologous island groups, with each group found in two or more strains. Additional 62 “orphan” islands were each found in a single strain (Appendix A).

The distribution of the 56 homologous island groups may inform us of the evolutionary lineages of *A. actinomycetemcomitans* (Figure 1). Hierarchical clustering analysis of the distribution of the islands identified four clusters. The results agree with the five clades identified in our phylogenetic analysis based on the core genes [3]. Here, the two closely related clades a/d and e/f were merged into a single cluster. The results suggest that acquiring genomic islands may have been a driving force in the evolutionary divergence of *A. actinomycetemcomitans.*

We noted distinct patterns of island acquisition (hereafter, it will be called PIA) among the *A. actinomycetemcomitans* strains. Some genomic islands were found in nearly all of the groups (Figure 1, PIA 1). These islands were likely present in the ancestral strain of *A. actinomycetemcomitans* and conferred critical functions to the bacteria; therefore, they were retained after the divergence of the species. Several genomic islands were missing in clade e’ (Figure 1, PIA 2), including the well-known cytolethal distending toxin island (see later section for more details). The presence of these island families in all strains except for the clade e’ strains is consistent with our previous study, suggesting an early divergence between clade e’ and the other clades [3]. Several homologous island families (Figure 1, PIA 3–6) appear to be clade-specific. For example, PIA 3 was found in clades a/d/e/f, PIA 4 was found in clade b, PIA 5 was found in clade c, and PIA 6 was found in clade e’. These islands may be retained because of their critical roles in adapting to niches unique to each clade.

Our grouping of *A. actinomycetemcomitans* in this study and previous studies shares some similarities with two other phylogenetic studies of *A. actinomycetemcomitans* [3,4,9,10]. Jorth and Whiteley divided 17 strains of *A. actinomycetemcomitans* into three lineages by phylogenetic analysis of 30 concatenated core genes [9]. They further examined the natural competence and the clustered regularly interspaced short palindromic repeats (CRISPRs) among the strains. They concluded that the competence loss, followed by the loss of CRISPRs, played a role in the emergence of different lineages of *A. actinomycetemcomitans*. Lineage 1 of their study comprised a single strain, which corresponded to our clade e’. Lineage 2 and lineage 3 corresponded to our clades b/c and clades a/d/e/f, respectively. Nedergaard et al. examined 35 strains of *A. actinomycetemcomitans* by whole genome sequencing and identified three lineages [10]. Lineage 1 corresponded to our clades b/c. Lineage 2 included strains of clades a/d/e/f and a serotype g strain, which was not included in our study. Interestingly, Nedergaard et al. identified lineage 3 comprising serotype a strains distinct from the serotype a strains in lineage 2 [10]. Also, in their study, two serotype e strains, designated as clade e’, were the outgroup in the phylogenetic analysis. This study also confirmed the distinction of clade e’ strains from other *A. actinomycetemcomitans* lineages. The clade e’ strains were first described by van der Reijden et al. [11]. They identified a group of serotype e strains that differed from other *A. actinomycetemcomitans* strains in the 16S rDNA gene sequences, amplified fragment length polymorphism typing, and carbohydrate fermentation analysis [11]. Reijden et al. designated these strains as serotype e’, representing a distinct subgroup within the species based on DNA–DNA hybridization analysis [11]. However, based on average nucleotide identity, these clade e’ strains appear to represent a different species in the genus *Aggregatibacter,* as proposed by Nedergaard et al. [12]. Similarly, the strain RHAA1 and similar strains from rhesus monkeys should be assigned to a distinct species based on their phylogenetic distance from other *A. actinomycetemcomitans* clades.

Above, we show that on a large scale, island inheritance patterns follow a similar inheritance pattern as sequence variation within the clades, e.g., phylogenetic trees constructed from sequence variation in core genes are similar to dendrogram trees constructed from the presence or absence of the islands. On a fine scale, e.g., at the level of individual island groups, the patterns of inheritance are far more complex. While a given genomic island may be common within a clade, the structure of the island often varies slightly between the members of the same clade. For example, there may be a small deletion, insertion, or inversion within an island in one isolate that is not present on the same island in other isolates in the same clade.

A detailed analysis of the inheritance patterns of all islands is beyond the scope of this work and likely beyond the interest level of most readers. Below, we highlight the observations of the three island groups: the leukotoxin operon, the cytolethal distending toxins, and the serotype-specific gene cluster. These three island groups were selected because they are important virulence determinants and demonstrate a wide range of inheritance patterns and mechanisms observed within an island group.

### 2.3. Acquisition of Leukotoxin Operon Marked the Distinction between A. actinomycetemcomitans and A. aphrophilus

*A. actinomycetemcomitans* and *A. aphrophilus* exhibit a high degree of similarity in both nucleotide sequence and gene content [3]. These two species share similar oral niches but differ in their potential to cause periodontal diseases [13,14,15]. Therefore, the distinct evolutionary pathways are of great interest for understanding the microbial pathogenesis of periodontitis. From the 2648 *A. actinomycetemcomitans* core genes, 2264 (85.5%) were found in *A. aphrophilus*. The remaining 384 (14.5%) *A. actinomycetemcomitans* core genes, which are not found in *A. aphrophilus*, are scattered randomly as isolated genes across the genome except for the leukotoxin operon, a major virulence determinant of *A. actinomycetemcomitans* [15,16]. We found that the leukotoxin operon is the only large-sized core genomic region (>4000 bp) missing in *A. aphrophilus*. Figure 2 summarizes the different organizations of the locus in the strains of these two species.

The most common configuration of the locus, found in the strains of all genetic clades, is the largest island that includes the leukotoxin operon and six flanking genes. The next largest island was found in a few serotype b strains (RHAA1, ANH9381, I23C, and S23A) that showed a deletion of the three downstream genes starting from the gene encoding the PTS system mannitol-specific transporter subunit IICBA. The JP2 strain serotype b HK1651 contains the smallest island with an additional truncation of the upstream serine hydroxymethyltransferase. It seems likely that the longer construct of the locus in *A. actinomycetemcomitans*, which is found in most strains, was the parental construct that underwent deletions, leading to two variations found in some but not all serotype b strains.

The locus in *A. aphrophilus* NJ8700 revealed the homologous genes encoding serine hydroxymethyltransferase and oxidoreductase. However, the locus includes two genes for hypothetical proteins instead of the leukotoxin operon. The locus in *A. aphrophilus* ATCC33389 revealed a deletion of the two genes encoding hypothetical proteins.

The analyses suggest that the acquisition of the leukotoxin operon may be a significant evolutionary event that led to the speciation of *A. actinomycetemcomitans* and *A. aphrophilus* during a descent from a common ancestor. The leukotoxin island acquired by the *A. actinomycetemcomitans* strains underwent a further reduction in some clade b strains. Notably, the smallest leukotoxin island was found in the high-leukotoxic JP2 strain HK1651, suggesting that the deletion in the promoter area of the leukotoxin operon may confer an advantage for these strains known to be associated with aggressive periodontitis [2,17].

### 2.4. Cytolethal Distending Toxin Operon Was Independently Acquired in Different A. actinomycetemcomitans Strains

The cytolethal distending toxin operon is a major virulence determinant of *A. actinomycetemcomitans* [15,18,19]. The genes encoding the toxins reside on genomic islands (cdt-islands) that are structurally heterogeneous in different *A. actinomycetemcomitans* strains [4,20]. This study further revealed the complexity of the cdt-islands in *A. actinomycetemcomitans*. The various genetic constructs of the cdt-islands and their flanking regions are depicted in Figure 3. The cdt-island may be found in one of two loci, as shown in Figure 3a (henceforth, they will be designated as cdt-locus I and cdt-locus II). The cdt-locus I is flanked by the genes that encode tRNA (next to CDP-diacylglycerol-glycerol-3-phosphate 3-phosphatidyltransferase) and NAD-dependent deacetylase, as shown in Figure 3b. The cdt-locus II is flanked by genes encoding Dca and tRNA (next to oligoribonuclease) (Figure 3c).

The cdt-islands were identified in the cdt-locus I in the clade b and clade c strains (Figure 3b), while the cdt-islands were found in the cdt-locus II in the clade a/d and e/f strains (Figure 3c). Additional differences in the cdt-islands were also noted. In particular, the cdt-islands of the clade b and c strains exhibited significant variations in size, possibly due to large-scale gene gains or losses (Figure 3b). On the other hand, the cdt-islands in the clade a/d and e/f strains were similar in size at 15.2 kbp (Figure 3c). Notably, two clade e’ strains showed distinct structures at the cdt-locus I. The strain SC936 of clade e’ showed a large non-cdt-genomic island, while other clade e’s strains did not have any insertions at the locus.

The genomic islands found in the cdt-locus I and cdt-locus II have features that suggest distinct acquisition mechanisms. The cdt-islands in the cdt-locus I included genes that encode transposase (closely related to a plasmid). On the other hand, the cdt-islands in the cdt-locus II included genes that encode conjugal transfer proteins, relaxase, and integrase, closely related to phages.

We speculated that the cdt-free cdt-loci represented the ancestral forms. The cdt-island was then acquired independently by either a plasmid-mediated mechanism (to the cdt-locus I) or a phage-mediated mechanism (to the cdt-locus II). The cdt-island in the cdt-locus I underwent additional modifications characterized by gene deletions. The hypothesis also suggested an early evolutionary divergence between clades b/c and clades a/d/e/f, consistent with the conclusion from our previous study.

### 2.5. Serotype-Specific Gene Clusters Are Found in Two Different Loci

The serotype-specific antigen gene clusters for *A. actinomycetemcomitans* have been of considerable interest due to their action in producing one of the immunodominant antigens of *A. actinomycetemcomitans* and a potential marker to identify highly virulent strains in the species [14,21,22,23,24,25,26,27,28]. To date, *A. actinomycetemcomitans* isolates have been divided into the following seven serotypes: a, b, c, d, e, f, and g, which are based on their surface carbohydrate antigens and/or the genetic loci that encode the genes responsible for the synthesis of these antigens. Given the sequence and gene content variability observed in the serotype-specific loci, strains assigned to the same serotypes based on serotype-specific gene clusters may have slightly different surface antigens.

The phylogenetic tree of *A. actinomycetemcomitans*, which is based on concatenated core genes in our previous study, suggests that clade e’ split early from the ancestral lineage [3]. The remaining clades, including serotype a–f strains, are further divided into clades a/d/e/f in one branch and clades b/c in another. Subsequent evolution led to clades a/d splitting off from clades e/f and clade c and clade b splitting off from each other [3].

Here, we examined serotype-specific loci using a representative genome for each clade. This approach captures much of the evolution of the serotype-specific traits within a clade but ignores the potential variability of the surface antigen within a clade. Notably, we found that the serotype-specific antigen gene cluster’s evolutionary history does not conform to the core gene-based phylogenetic tree.

Figure 4 summarizes the serotype-specific gene cluster loci using a representative strain for each clade. In each *A. actinomycetemcomitans* genome, one of the two identified loci (namely locus I and locus II) harbors its serotype-specific gene cluster, as shown in the genomic maps of serotype a D7S-1 and serotype c D11S-1 (Figure 4a). While the serotype a-specific gene cluster was found in locus I, the serotype-specific gene clusters of all other serotypes were found in locus II.

The serotype-specific antigen locus for serotype a consists of six variable genes (marked with asterisks) plus five core genes. The gene cluster is flanked by the major facilitator transporter and a transcriptional regulator. As shown previously, the serotype a-specific polysaccharide antigen of *A. actinomycetemcomitans* is composed of the unusual sugar 6-deoxy-d-talose [29,30]. The two genes encoding GDP-alpha-d-mannose 4,6-dehydratase and GDP-4-keto-6-deoxy-d-mannose reductase, which are found in this locus, are necessary for the biosynthesis of (GDP)-6-deoxy-d-talose, the activated form of 6-deoxy-d-talose [31].

The organization of the gene clusters for serotype b–f strains in locus II is depicted in Figure 4c. Instead of providing a detailed discussion of their organizations among the clades, we highlight the observations that are relevant to the evolution of *A. actinomycetemcomitans.* Locus II of serotypes b–f contains a block of the following four genes: *rmlA, rmlB, rmlC*, and *rmlD* genes. These genes are necessary for the production of dTDP-L-rhamnose, but L-rhamnose is not present in the O-polysaccharides of LPS in serotype c [25,32,33,34,35]. A second *rmlB* homolog (referred to as *rmlB-2*) was also found in locus II in all serotypes except for the serotype d strain, including serotype a. In Figure 4c, the *rmlB* in the strains of serotypes b, c, e, and f on the left (immediately adjacent to lytic transferase) is most similar in sequence to the single copy found in serotype d. We shall refer to this gene as *rmlB-1* and the other as *rmlB-2*. The *rmlB-2* gene (or non-functional homologs) of serotypes b, c, e, and f is most similar to the single copy found in isolates of serotype a. Therefore, clade a/d strains contain only one copy of the *rmlB* gene (*rmlB-2* in serotype a and *rmlB-1* in serotype d). In contrast, the strains of serotypes b, c, e (including the genetically distinct clade e’), and f contain either two intact copies of this gene or one intact copy of *rmlB* and one that is no longer functional.

Finally, the genes in the serotype-specific locus of clades e and e’ are most closely related. This suggests that these genes were acquired after the split defined in the core gene-based phylogenetic tree. An intact IS-200-like transposase is found in this locus in clade e and c strains but not in the other serotype strains and, in particular, not in the clade e’ strains. This suggests that this locus or parts of this locus were acquired as a result of transposition in at least some of the strains.

To examine the evolution of the *rmlB* genes, we performed a phylogenetic analysis of all *rmlB* sequences and homologous regions of multiple fully sequenced isolates (Figure 5) and designated the clades as serotype-specific clades (s-clade) to distinguish them from the clades based on the core genes.

The *rmlB-2* gene is more similar to the *rmlB* gene found in the *A. aphrophilus* genomes, which suggests it is the more ancient copy. Interestingly, both *rmlB-1* and *rmlB-2* are found in the rhesus macaque isolate of *A. actinomycetemcomitans* (RHAA1). This suggests either that the second copy of *rmlB* was inherited independently in the human and rhesus strains or, more likely, that the two copies of *rmlB* existed in the *A. actinomy**cetemcomitans* before the split of the human and rhesus lineages.

The *rmlB-2* gene in the s-clade e’ is similar to the rhesus *rmlB*-2 gene (98.67%) as shown in Appendix A. The *rmlB-2* gene of serotype b, c, and e formed a cluster. The sequence identity within the *rmlB-2* s-clade b/c/e branch is 99.59–100%, whereas the sequence identity between the *rmlB-2* s-clade b/c/e clade and the s-clade e’/rhesus clade is 90.45–90.86%. The *rmlB-1* genes within the s-clade e/e’ are highly similar (99.79–100%) and similar to the *rmlB-1* gene in the rhesus isolate (98.46–98.56%). The *rmlB-1* genes within the s-clade b/d/f are highly similar (99.18–100%) but are only 94.46–94.87% similar to the s-clade e/e’ and the rhesus *rmlB*-1 genes. The results show that the evolutionary history of both *rmlB* genes differs from the history inferred from the core gene-based phylogenetic tree.

## 3. Materials and Methods

### 3.1. Identification of Core Genes, Accessory Genes, and Genomic Islands

Appendix A lists the 32 *A. actinomycetemcomitans* strains used in this study. The genome sequences were obtained from the NCBI database. To identify the core and variable genes, all genes were first compared and grouped into homologous gene clusters based on their sequence similarity using our protocol, as previously described [3]. Briefly, the genes were compared at both the nucleotide and protein sequence levels and grouped into homologous gene clusters using a sequence similarity cutoff of at least 30% and an alignment coverage of query and hit sequences cutoff of at least 50%. The homologous gene cluster data was organized in the form of a “gene table”, in which the rows represent unique genes, the columns represent strains, and the present or absent status is indicated in the cell. The gene table was then analyzed to identify core (genes that are present in all genomes) and variable (genes that are missing in one or more genomes) genes. Poor or low-quality genome assemblies usually result in a lower rate of gene calling. Hence, to reduce false negatives in the core gene set, we ignored 9 genomes based on the following criteria: (i) Genome assembly with more than 500 contigs (strain SA508, SA3733, SA269, and ANH9776); (ii) clonal strains of the same individual (strain A160, SCC4092, S23A, and AAS4A); (iii) a non-human strain (strain RHAA1). This left us with 23 unique genomes for the human-associated *A. actinomycetemcomitans* strains. Genes that are present in all of these 23 genomes were classified as core, and genes that were present in only a subset were classified as variable genes. The core and variable classifications in the gene table were then mapped back to genes in all 32 genomes. Finally, the genomic islands in each genome were defined as regions of adjacent variable genes when the region is at least 4000 bases in length.

### 3.2. Grouping of Genomic Islands

To identify islands that are likely related, we grouped the genomic islands based on their gene content similarity. A custom Perl script was implemented to assign the genomic islands that share 80% or more homologous genes to the same island group. The island group data was then used to determine the distribution patterns of the genomic islands across the *A. actinomycetemcomitans* genomes. Finally, the relationship between the genomes based on their genomic island presence or absence profiles was visualized using the hierarchical clustering method within the MeV software, using the default settings [36].

### 3.3. Features of Island Insertion Sites

To identify the genomic island insertion sites, we identified the core genes adjacent to each genomic island in both the upstream and downstream directions. These pairs of core genes were then identified as genomic island insertion sites and used to compare with other genomes to understand the differences in the genomic islands found within these insertion sites. The presence of different islands at a common insertion site across different isolates would be an indication of an insertional hot spot. The visualization of the genomic island insertion sites across the genomes was performed using the EasyFig software [37]. Genomes where an insertion site spanned multiple contigs were omitted from each visualization.

## 4. Conclusions

In this study, we identified and analyzed the variable genes and genomic islands of *A. actinomycetemcomitans* to better understand their potential roles in the genetic divergence of *A. actinomycetemcomitans*. By analyzing the distribution patterns of the islands, our results show possible acquisition patterns and suggest the probable history of the genomic islands. For example, the genomic islands that are present in all major phylogenetic clades of *A. actinomycetemcomitans* may be more ancient compared to the ones that are present in subsets of clades. The clade-specific genomic islands identified in this study can be useful targets for further studies to understand their distinct roles in each clade. The acquisition of the leukotoxin operon might be a major evolutionary event that led to the speciation of *A. actinomycetemcomitans* and *A. aphrophilus* during the descent from a common ancestor, as this is the only large core genomic region of *A. actinomycetemcomitans* that is not present in *A. aphrophilus*. In addition, the analysis of this locus showed that the leukotoxin region had undergone further reduction that confers advantages for various strains. Finally, our analysis of the cytolethal distending toxin and the serotype-specific gene cluster islands showed that while these islands were found in multiple strains, their inheritance may differ as they can be found in different positions in the genome. Overall, the results demonstrate possible variations in island inheritance patterns and provide clues on the mechanisms that introduce these islands into the genomes of *A. actinomycetemcomitans*.

## Figures and Tables

**Figure 1 pathogens-11-01037-f001:**
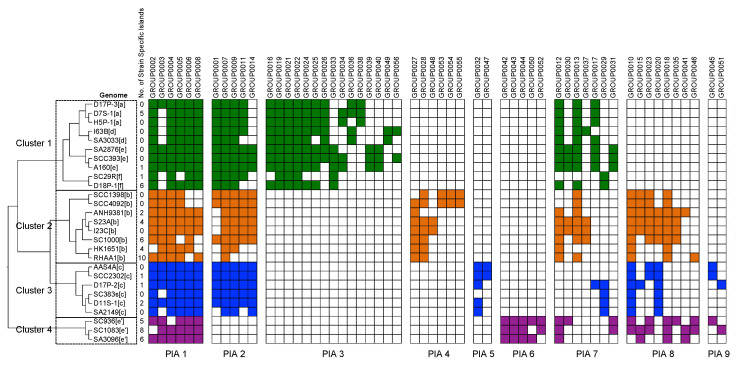
Patterns of island acquisition (PIA) among the strains. PIA 1, islands found in all clusters; PIA 2, islands found in cluster 1 (clades a/d and e/f), 2 (clade b), and 3 (clade c); PIA 3, islands specific to cluster 1; PIA 4, islands specific to cluster 2; PIA 5, islands specific to cluster 3; PIA 6, islands specific to cluster 4 (clade e’); PIA 7 is similar to PIA 1 but it is less represented in clusters 2, 3, and 4; PIA 8 is absent in cluster 1; PIA 9 is absent in clusters 1 and 2.

**Figure 2 pathogens-11-01037-f002:**
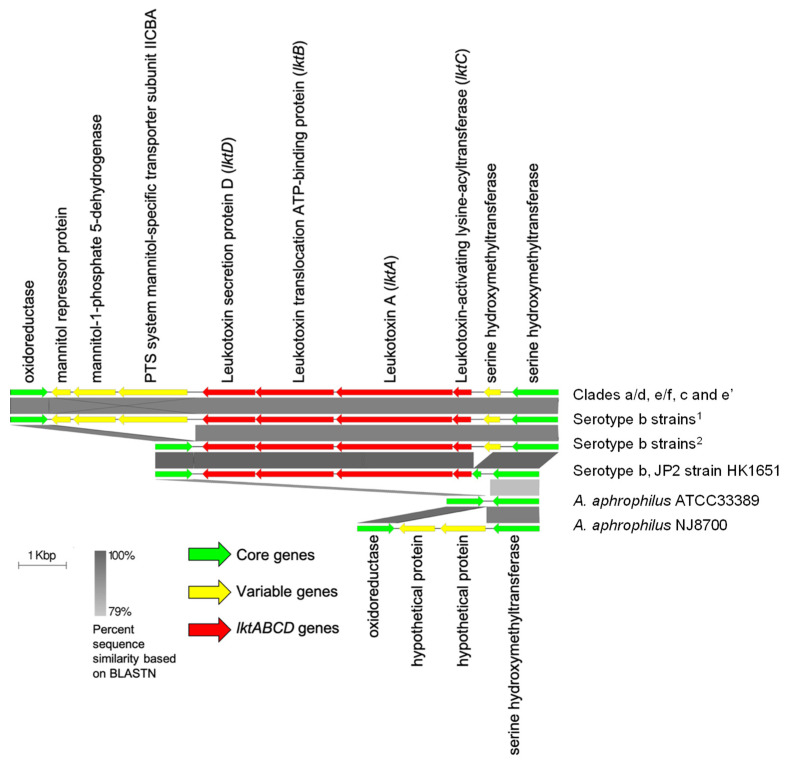
The leukotoxin operon of *A. actinomycetemcomitans*. The leukotoxin operon (the *lktABCD* genes are shown as red arrows) is found between the genes encoding oxidoreductase and serine hydroxymethyltransferase in all of the *A. actinomycetemcomitans* genomes. Even though similar regions containing genes encoding oxidoreductase and serine hydroxymethyltransferase are present in the *A. aphrophilus* genomes, the leukotoxin operon is not found in this locus or elsewhere within the *A. aphrophilus* genomes. Clades a/d, e/f, c, and e’ include D17P-3, D7S-1, I63B, SA3033, D18P-1, SA2876, SCC393, A160, SC936, SC1083, D11S-1, D17P-2, SC383s, SCC2302, and AAS4A. ^1^ Serotype b strains include RHAA1, ANH9381, I23C, and S23A. ^2^ Serotype b strains include SC1000, SCC1398, and SCC4092.

**Figure 3 pathogens-11-01037-f003:**
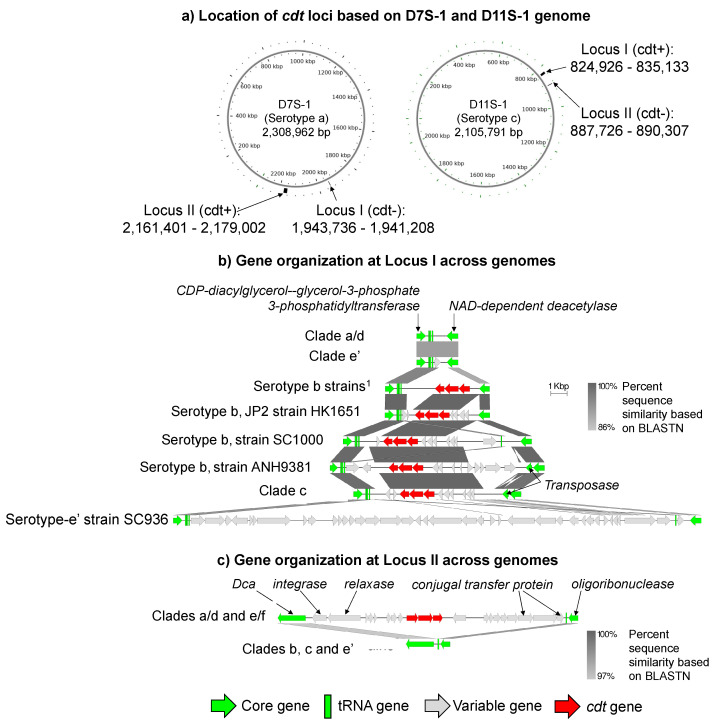
Location and features of the cdt-island among the *A. actinomycetemcomitans* strains. The cytolethal distending toxin operons appear to have been independently acquired in the clade b/c strains relative to the clade a/d and e/f/ strains as they are found in two different genetic loci (the cdt-locus I in clades b and c and the cdt-locus II in clades a/d and e/f). (**a**) Locations of the cdt-locus I and cdt-locus II in D7S-1 (clade a/d, serotype a) and D11S-1 (clade c, serotype c) genomes. The circular genomes are displayed so that the origin of replication is at the zero o’clock position. The cdt-locus I is characterized by core genes that encode CDP-diacylglycerol--glycerol-3-phosphate 3-phosphatidyltransferase and NAD-dependent deacetylase. On the other hand, the cdt-locus II is characterized by core genes that encode Dca and oligoribonuclease. (**b**) Overview of the cdt-locus I across the genomes. Clade a/d strains are D17P-3, D7S-1, H5P-1, and SA3033. Clade e’ strains are SC1083 and SA3096. Serotype-b strains 1 are SCC1398 and SCC4092. Clade c strains are D11S-1, D17P-2, SCC2302, and AAS4A. (**c**) Overview of the cdt-locus II across the genomes. Clades a/d/e/f are D17P-3, D7S-1, H5P-1, I63B, D18P-1, SA2876, SCC393, and A160. Clades b, c, and e’ strains are SC936, SC1083, RHAA1, HK1651, ANH9381, SCC1398, SCC4092, I23C, S23A, D11S-1, D17P-2, SC383s, SCC2302, and AAS4A.

**Figure 4 pathogens-11-01037-f004:**
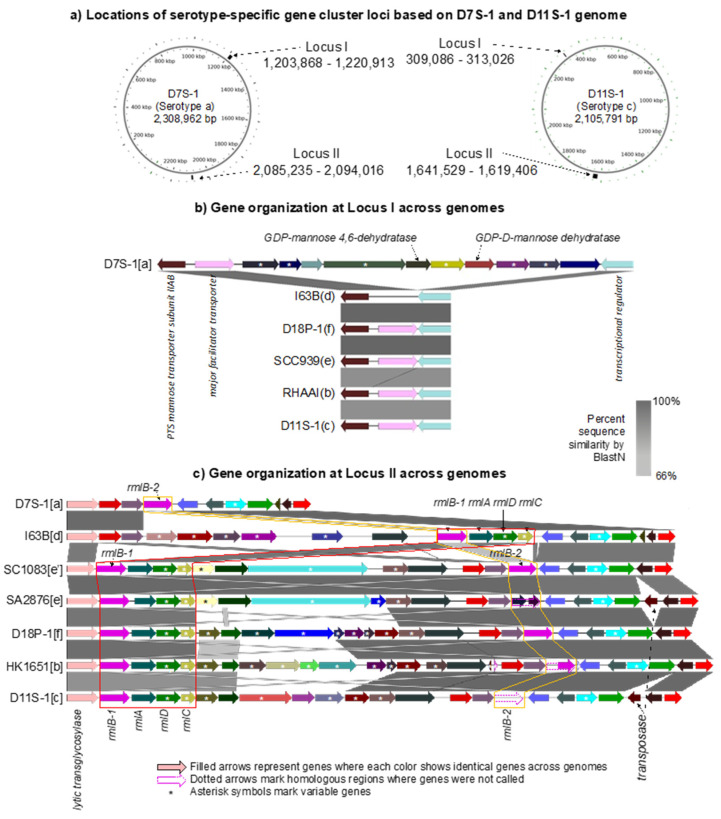
The serotype-specific gene cluster of *A. actinomycetemcomitans*. (**a**) Two different loci that contain serotype-specific gene clusters. Locus I start and stop coordinates from D7S-1. Locus II start and stop coordinates from D11S-1. (**b**) Locus I only contains a serotype-specific gene cluster in the serotype a strains. The gene cluster is shown with D7S-1 as the exemplar. Other serotype a strains with a similar structure in locus I include D17P-3, D7S-1, and H5P-1. Examples of locus I in other serotypes are shown below D7S-1. (**c**) The serotype-specific gene cluster in locus II is shown for each of the seven representative strains. The serotype of the strain is provided in parenthesis.

**Figure 5 pathogens-11-01037-f005:**
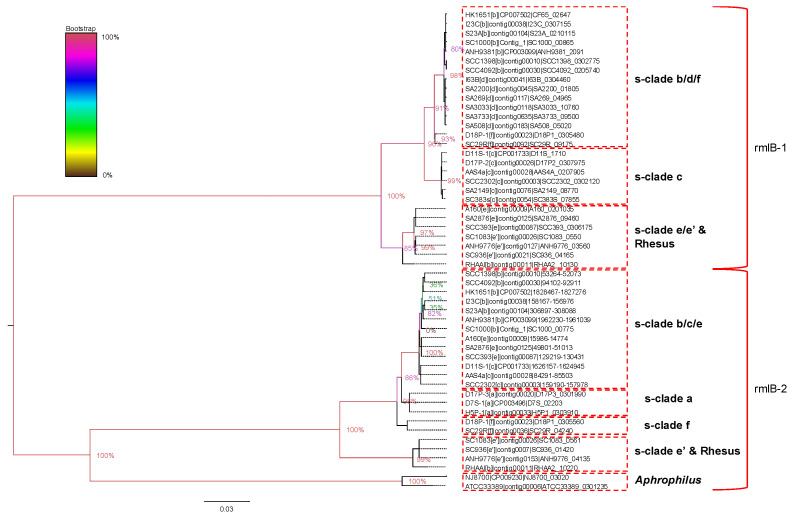
Phylogenetic analysis of all rmlB sequences and homologous regions. The nucleotide sequences of the *rml**B* genes (*rmlB-1* and *rmlB-2*) and the homologous genomic regions were used to create a phylogenetic tree to summarize their evolutionary relationship. The *rml**B* gene sequences from *A. aphrophilus* were also included. The name for each sequence shows the genome name followed by contigID and followed by either the geneID or the start–stop coordinates of the homologous genomic region.

## Data Availability

The data can be accessed via our online Gene Table Viewer tool at http://expression.washington.edu/genetable/script/gene_table_viewer?organism=aa_ha&build=17_07_20 (accessed on 21 July 2022).

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
