# Peer review of "Genomic Islands Shape the Genetic Background of Both JP2 and Non-JP2 Aggregatibacter actinomycetemcomitans"

_pathogens, 2022, doi:10.3390/pathogens11091037_

Round 1
Reviewer 1 Report
Pathogens-1852842, Genomic islands of Aggregatibacter actinomycetemcomitans
The manuscript (MS) investigates whole genome sequences (WGS) of Aggregatibacter actinomycetemcomitans (AA) with special focus on variable genes and genomic islands. By use of more “lenient” criteria to define core genes (section 2.1), it was possible to categorise 62% of unique gene clusters as present in all strains, and restricting variable genes to the reminding 1,628 unique genes. The main part of the MS thereafter focuses on observations of three selected island groups: the leukotoxin operon, the cytolethal distending toxin and the serotype-specific gene cluster. Introduction of another species (Aggragatibacter aphrophilus) makes possible the description of the leukotoxin operon as a genomic island, because leukotoxin operon (present in all strains of AA) is the only large-size core genomic region (>4,000 bp) missing in A. aphrophilus. The cdt operon is present in two different loci in all strains of AA sensu stricto, but absent from serotype e’. The serotype-specific genes are also present in two different loci, locus 1 for the serotype a strains, and locus II for all other serotypes.
Overall comments.
The MS provides an in-depth characterisation of previously examined WGS of AA, with focus on phenotypic and clinical important operons. My major concern with the MS is the selection of WGS to be included, which influences the definition of the species and the characterisation of the population structure. It is difficult for the reader to grasp the difference between 32 strains (all), 23 unique genomes (section 3.1), or 27 WGS (after exclusion, section 2.2). Apparently, serotype e’ and rhesus monkey strain RHAA1 are included in the present description of the species, which is in contrast to other descriptions (Jorth & Whiteley, An Evolutionary Link between Natural Transformation and CRISPR Adaptive Immunity. MBio 2012). After delineation of the species, the population structure can be addressed, and the authors refer to several classifications. Serotypes (a-g) are based on outer membrane O polysaccharides, and bears importance because of history and general acceptance. Previously, the authors defined five clades (a/d, e/f, b, c, and e’ (Kittichotirat et al 2011). In the current Figure 1, this is reduced to four clusters by combining a/d and e/f. In figure 3, strains are divided in two groups, b+c (locus I) and a, d, e +f (locus II), which resembles lineage 2 or 3 by Jorth & Whitley, or lineage I or II by Nedergaard (Nedergaard et al. Whole Genome Sequencing of Aggregatibacter actinomycetemcomitans Cultured from Blood Stream Infections, Pathogens 2019, 8, 256). WGS of strains of the Novel Lineage Expressing Serotype a Membrane O Polysaccharide, described by Nedergaard et al 2019, have not been examined by the authors.
It is mentioned in the text, that “Clades b/c strains from humans were smaller in genome size than other clades (Supplementary Table S1), suggesting genome reduction that accompanied their adaptation to humans”. The different explanation, suggested by Jorth & Whitely, that genome sizes are linked to presence or absence of competence, should be discussed.
Author Response
Critique: My major concern with the MS is the selection of WGS to be included, which influences the definition of the species and the characterisation of the population structure. It is difficult for the reader to grasp the difference between 32 strains (all), 23 unique genomes (section 3.1), or 27 WGS (after exclusion, section 2.2).
Response: Agree. We have extensively revised the results in section 2.1. Features of A. actinomycetemcomitans genomes. Here we simplified the description of the genomes, deleted Supplementary and allow the readers to focus on more interesting findings in other sections.
Critique: Apparently, serotype e’ and rhesus monkey strain RHAA1 are included in the present description of the species, which is in contrast to other descriptions (Jorth & Whiteley, An Evolutionary Link between Natural Transformation and CRISPR Adaptive Immunity. MBio 2012). After delineation of the species, the population structure can be addressed, and the authors refer to several classifications. Serotypes (a-g) are based on outer membrane O polysaccharides, and bears importance because of history and general acceptance. Previously, the authors defined five clades (a/d, e/f, b, c, and e’ (Kittichotirat et al 2011). In the current Figure 1, this is reduced to four clusters by combining a/d and e/f. In figure 3, strains are divided in two groups, b+c (locus I) and a, d, e +f (locus II), which resembles lineage 2 or 3 by Jorth & Whitley, or lineage I or II by Nedergaard (Nedergaard et al. Whole Genome Sequencing of Aggregatibacter actinomycetemcomitans Cultured from Blood Stream Infections, Pathogens 2019, 8, 256). WGS of strains of the Novel Lineage Expressing Serotype a Membrane O Polysaccharide, described by Nedergaard et al 2019, have not been examined by the authors.
It is mentioned in the text, that “Clades b/c strains from humans were smaller in genome size than other clades (Supplementary Table S1), suggesting genome reduction that accompanied their adaptation to humans”. The different explanation, suggested by Jorth & Whitely, that genome sizes are linked to presence or absence of competence, should be discussed.
Response: Agree. The revised sections 2.1 and 2.2 now include discussions of studies by Nedergaard et al. and Jorth & Whiteley. Specifically, we compared our results with the lineages of A. actinomycetemomitans identified in these two studies, noted the possible role of natural competence and CRISPRs in the evolution of the species, the distinction of clade e’ strains from other strains and the evidence for designating a new species for clade e’ strains.
Reviewer 2 Report
Dear authors,
In the study “Genomic islands shape the genetic background of both JP2 and non-JP2 Aggregatibacter actinomycetemcomitans” you investigated genetic distinctions of Aggregatibacter actinomycetemcomitans strains with regard to evolutionary background. For this purpose, you analyzed the sequences of 32 A. actinomycetemcomitans strains taken from NCBI. In addition, you identified the genetic basis for distinguishing between A. actinomycetemcomitans strains and A. aphrophilus strains. This plays an important role in characterizing possible genetic peculiarities that enable the distinction between periodontal pathogens and non-periodontal pathogens.
The manuscript is fully in line with the focus of the journal and the topic is of great importance in terms of personalized medicine.
The manuscript is clearly written, well-structured and the data are well presented. The technical methods used are appropriate.
In summary, this manuscript extends insight into evolutionary genetic features of A. actinomycetemcomitans strains and displays the knowledge in this area very concisely.
Author Response
We thank the reviewer for the positive comments. We have further improved the manuscript by addressing the critiques from the other reviewer.